# Determinants of intention to improve oral hygiene behavior among students based on the theory of planned behavior: A structural equation modelling analysis

Kegnie Shitu[1]*, Mekuriaw Alemayehu[2], Yvonne A. B. Buunk-Werkhoven[3], Simegnew Handebo[1]

1 Department of Health Education and Behavioral Sciences, Institute of Public Health, University of Gondar, Gondar, Ethiopia, 2 Department of Environmental and Occupational Health and Safety, Institute of Public Health, University of Gondar, Gondar, Ethiopia, 3 SPOH ARTS–International Oral Health Psychology, Amsterdam, The Netherlands

* kegnsh@gmail.com

**Data Availability Statement:** The datasets generated and/or analyzed in the current study are

## Abstract

### Introduction

The prevalence of oral hygiene behaviors (OHB) is very low among school children in Ethiopia. However, the determinants of student's readiness/intention to perform those behaviors have been remained unstudied.

### Objective

This study aimed to identify the determinants of oral hygiene behavioral intention (OHBI) among preparatory school students based on the theory of planned behavior (TPB).

### Methods and materials

An institution-based cross-sectional study was conducted among 393 students. A 98-item self-administered questionnaire was used to evaluate oral hygiene knowledge (OHK), oral hygiene behavior (OHB), and OHBI based on TPB variables [attitude (ATT), subjective norms (SN) and perceived behavioral control (PBC)]. Descriptive statistics and structural equation modeling analysis (SEM) were employed to confirm relationships and associations among study variables. A p-value of less than 0.05 and a 95% confidence interval were used to declare statistical significance.

### Results

A total of 393 students were participated with a response rate of 97.5%. The mean age of the participants (54% females) was 18 (± 1.3) with an age range of 16 to 24. The TPB model was well fitted to the data and explained 66% of the variance in intention. ATT (β = 0.38; 95% CI, (0.21, 0.64)), SN (β = 0.33; 95% CI, (0.05, 0.83)) and PBC (β = 0.29; 95% CI, (0.13, 0.64)) were significant predictors of OHBI, where ATT was the strongest predictor of OHBI.

publicly available at Kaggle [https://www.kaggle.com/kegnieshitu/determinants-of-ohbi-among-students-tpb].

**Funding:** KS received monetary fund for this research work. The fund was obtained from the University of Gondar with a grant code of UOGF/6417. Its organizational address is www.uog.edu.et. The funders had no role in study design, data collection and analysis, decision to publish, or preparation of the manuscript.

**Competing interests:** The authors have declared that no competing interests exist.

**Abbreviations:** AMOS, Analysis of a Moment Structures; AGFI, Adjusted Goodness of Fit Index; ATT, Attitude; GFI, Goodness of Fit Index; NFI, Normed Fit Index; OHB, Oral Hygiene Behavior; OHK, Oral Hygiene Knowledge; PNFI, Parsimony Normed Fit index; PBC, Perceived Behavioral Control; SEM, Structural Equation Modelling; SN, Subjective Norm; SPSS, Statistical Package for Social Science; SRMR, Standardized Root Mean Residual; TPB, Theory of Planned Behavior; WHO, World Health Organization.

## Conclusion

The TPB model explained a large variance in the intention of students to improve their OHB. All TPB variables were significantly and positively linked to stronger intent, as the theory suggests. Furthermore, these results suggest that the model could provide a framework for oral hygiene promotion interventions in the study area. Indeed, these interventions should focus on changing the attitudes of students towards OHB, creation of positive social pressure, and enabling students to control over OHB barriers.

## Introduction

Untreated caries in permanent teeth is the most prevalent oral condition affecting 2.5 billion people worldwide. Globally, 60–90% of school children are affected by dental caries [1]. In Ethiopia, oral disorders are becoming high related to increasing risky oral health behaviors where dental caries, and periodontal diseases, are the commonest of oral health disorders in the country [2–6]. Poor oral hygiene is one of the known behavioral risk factors for oral health disorders, which is very common among economically disadvantaged society [7]. On the other hand, the significant burden of oral health problems can be mitigated by adequate oral hygiene behavior (OHB) [8–10]. For instance, regularly brushing teeth with toothpaste twice a day and daily flossing are effective in preventing oral health problems like tooth decay and periodontal disease [11, 12]. Nonetheless, the prevalence of OHB among school children is extremely low in Ethiopia, usually, less than 10% where there is increasing in the incidence of risky oral health behaviors, such as high sugary food consumption and carbonated soft drinks following unplanned socioeconomic changes in the country [3, 5, 13–16].

The TPB is a renowned theory which was developed by Icek Ajzen as an attempt to predict how humans were perceived to perform several behaviors under the influence of intention [17]. According to this theory, human behaviors are to a large extent determined by the intention to perform that behavior. In turn, the behavioral intention is determined by three cognitive variables: ATT towards the behavior SN and PBC [18]. The theory has shown its utility in predicting various health behaviors. A meta-analysis has shown that the TPB accounted for 39% of the variance in intentions and 27% of the variance in behavior across a broad spectrum of behaviors [19]. In Ethiopia, the theory has been used to predict different health behaviors such as screening behavior [20], condom use [21], blood donation intention [22], and HIV risk behaviors [23].

The TPB has been found to be effective in predicting oral health-related intentions (OHBI) and behaviors across different populations, places, and time by different studies [24–29]. For example, this theory explained 52% of the variance in OHBI in a study conducted among Romanian students [24]. In another study done in the Dominican Republic, the TPB variables explained about 32% of the variance in intention to improve OHB [30]. Alternatively, the theory was also predicted about 64% of the variance in OHBI in a study done in Norway [27].

Furthermore, the variables in the theory were found to be strong predictors of oral hygiene intention/behavior by different studies in different ways. For example, ATT, SN and PBC according to studies in Indonesia, Pennsylvania and Northern Ireland [29, 31, 32], ATT and PBC based on studies in Ireland, Canada and Romania [24, 28, 33], and SN according to a study conducted in Iran [34] were found to be a significant predictor(s) of oral hygiene behavior/intention.

Even though TPB is an effective framework for predicting oral health behaviors [35], no studies have been conducted in Ethiopia on the application of this theory to predict

intentions/behaviors related to oral health. Furthermore, activities to promote oral hygiene in schools are overlooked in the country's education system. If the opposite were to happen, identifying the psychosocial determinants of OHBI would be of paramount importance in the design and implementation of behavioral change interventions in oral hygiene among students [35]. However, little is known of such predictors in the study area or in the country in general. Therefore, the objective of this study was to produce preliminary evidence regarding the determinants of OHBI based on the TPB framework (S1 Fig).

## Methods and materials

### Study area and period

The institutional-based cross-sectional study design was conducted among preparatory school students who were attending their class at the selected private and public preparatory schools in Gondar city in the 2019/2020 academic year. Gondar city is located at about 727 Kilometers (KM) away from Addis Ababa, the capital city of Ethiopia, and 180 km away from Bahirdar the capital city of Amhara Regional State. Gondar city has a total area of 192.3 square KM with a total population of 338, 646. There are 10 (three private and 7 public) preparatory schools in the city administration. In these schools, there are a total of 7, 956 (4,143 females and 3813 males) students. Moreover, 856 and 7100 of them attend private and public preparatory schools respectively [36].

### Study participants

For this study, participants were preparatory school students attending Grade 11 and 12 of the 2019–20 academic year. In Ethiopia's education system, preparatory school refers to a post-secondary institution where students learn for two years prior to university entry. It's just a place where students are prepared to join an undergraduate degree.

### Inclusion and exclusion criteria

**Inclusion criteria.**   All students of the preparatory school of the city of Gondar in 2019/2020 have been included in this study.

**Exclusion criteria.**   Students from the preparatory school who were unavailable at the school during the data collection period were excluded after a home check. Furthermore, students who were transferred in/out during the academic year in which the study was carried out were also excluded.

### Sample size determination and sampling procedure

The minimum required sample size for the present study was calculated using the statistical calculator designed to sample size determination for a SEM, which was developed by the American professor, Dr. Daniel S. Soper in 2006 [37]. The calculation was based on the following assumptions; power = 80%, number of latent variables = 9, number of observed variables = 50, minimum anticipated effect = 0.3 (since there was no study done previously in Ethiopia), type one error ($\alpha$) = 0.05, design effect = 2 and non-response rate = 10%. Thus, the required sample size for the study was computed to be 403.

To recruit the required participants, a stratified multistage simple random sampling technique was employed. First, stratification was done based on school type into private and public/governmental schools, resulting in 3 and 7 private and government schools respectively. Secondly, three (one private and two public) preparatory schools were selected on a random basis. Then, 11 sections from public schools, and 2 sections from a private school were selected

randomly. Finally, students were selected randomly based on their class roaster using Microsoft excel random number generator.

## Study variables

In a multivariate analysis variable are classified in to four categories involving endogenous, exogenous, latent and observed variables. In this regard, the endogenous (dependent) variables of this study were intention (outcome variable), direct attitude, direct subjective norm, and direct perceived behavioral control. On the other hand, the indirect attitude, indirect subjective norm, indirect perceived behavioral control, self-reported OHB, OHK, age of the student and parental educational status were exogenous (independent) variables. All of the variables were unobserved (latent) except age and familial educational status.

## Data collection and measurement

The data was collected from march 2nd to 13th 2020 using a questionnaire which was developed based on an elicitation study and previous literatures [17, 18, 26, 29, 30, 38]. The instrument was initially prepared in English and then translated into the local language (Amharic) and translated back to English to check for its consistency. Content validity test and pre-test of the instrument were done based on seven experts and 21 preparatory school students respectively. Necessary amendments on the questionnaire were made upon the pertest and content validity results. The final questionnaire was composed of 98 items with four sections measuring socio-demographic, OHK, OHB, and TPB variables. Moreover, four BSc nurses and two public health professionals were participated as data collector and supervisor in the data collection process respectively after a one they received one-day training.

**Oral hygiene knowledge.** OHK was measured by 11 items having a true/false response category prepared based on the earlier Buunk-Werkhoven study [26]. Examples: "When my gum does not bleed while brushing my teeth, there is nothing wrong with my gum," and "For tooth care, it doesn't matter if we use our toothbrush for a long time unless it is broken or lost." Items were scored as correct = 1 and incorrect = 0, and the total score of OHK was computed by adding 11 items. The sum score ranged from 0–11, ($\alpha$ = 0.65).

**Self-reported oral hygiene behaviour.** The measurement of this section was also adapted from the OHB index used by Buunk-Werkhoven [26]. A culturally validated version of this OHB index included eight items concerning OHBs. The sum score of the index was in the range of 0–17. A higher sum score indicated better OHB, ($\alpha$ = 0.74).

Before the assessment of the TPB variables regarding oral hygiene behavior, the focal adequate OHB was described as "brushing your teeth twice a day (once after breakfast and once before going to sleep), using a soft-bristled toothbrush and fluoride-containing toothpaste; brushing softly∕without pressure for at least 2 min; brushing stepwise by making small strokes–sort of massage–near the gum, along the inside and the outside, and on the jackdaw areas. In addition to tooth brushing, daily interdental cleaning (i.e. use of floss, tooth sticks, or interdental brushes) and tongue cleaning are also recommended" [26].

## Direct measures of the TPB

**Attitude.** Direct ATT was measured by nine items that assessed the anticipated value of performing OHB regularly. Each item has a seven-point scale with 1 and 7 anchored by each end of the semantic differential. Participants were asked to show their position on how they evaluate the OHB, e.g., regular tooth brushing twice a day as described above is, 1 = unhealthy to 7 = healthy, 1 = unpleasant to 7 = pleasant, and so on. The total score ranged from 7 to 63 and a higher score indicates a favourable ATT towards OHB [39], ($\alpha$ = 0.89).

**Subjective norms.**    It is about the perceived social pressure by the participants concerning OHB and was measured by seven items having 7-point scales. Examples, "Most people who are important to me think that I should brush my teeth twice a day using toothpaste regularly as described above" and "It is expected of me that I brush my teeth twice per day twice a day using toothpaste regularly as described above" (1 = disagree to 7 = agree). The total score ranged from 7 to 49 the higher score indicates high social influence towards intention to OHB [39], ($\alpha$ = 0.79).

**Direct perceived behavioral control.**    It was assessed by four indicators, all measured by 7-point scales. Examples: "For me to brush my teeth twice per day using toothpaste regularly as described above is" (1 = difficult to 7 = easy), and "I am confident that if I wanted to, I could brush my teeth twice per day using toothpaste regularly as described above" (1 = false to 7 = true). The total score ranged from 4 to 28 and the higher score indicates the higher perceived ability of individuals to control factors to improve OHB [39], ($\alpha$ = 0.8).

## Indirect measures of the TPB

**Indirect attitude.**    The indirect attitude was measured based on four outcome evaluations and the corresponding four behavioral beliefs. Respondents were first required to indicate the likelihood that each outcome that would occur if they were engaged in oral hygiene behavior as recommended, for example, "If I brush my teeth, I will keep my teeth beautiful". They were then asked to evaluate each outcome, for example, "For me, having beautiful teeth is something important" on the agree/disagree dimension. Finally, behavioral beliefs were multiplied by the corresponding outcome evaluations, and then the summed product was used as the measure of indirect ATT. The composite score ranged from 6 to 196 and the higher score indicates a higher/favourable ATT towards OHB [18], ($\alpha$ = 0.89).

**Indirect subjective norm.**    An indirect measure of the SN was derived from the expectations and observations of five referents: parents, siblings, classmates, close friends, and teachers. Respondents were first asked to indicate the extent to which each of their significant others would endorse their intention to perform the recommended OHB. This was followed by a request to indicate the extent to which they were motivated to comply with the wishes of their significant others, across a seven-point semantic differential scale (1 = agree to 7 = agree). Each normative belief was multiplied by the corresponding motivation to comply and the summed product served as a measure of the indirect subjective norm. The composite score ranged from 20 to 490 and the higher score indicates the higher positive social pressure from significant others [18], ($\alpha$ = 0.92).

**Indirect perceived behavioral control.**    The indirect measure of PBC was grounded on the five beliefs elicited from the focus group discussions and in-depth interviews. It was measured based on control beliefs of participants i.e. a respondent's belief on the facilitators/barriers of oral hygiene behavior, for example, "How often do you face lack of toothpaste?" (1 = very rarely to 7 = very frequently) and the perceived power that they had to control those control beliefs, for example, "If I had faced lack of toothpaste, it would make it more difficult for me to brush my teeth twice a day by using toothpaste regularly", (1 = agree to 7 = disagree). The score of the variable was obtained in the same way to the indirect subjective norm and indirect attitude and its total score ranged from 10 to 245. A higher score indicates participants increased the power to control barriers to OHB [18], ($\alpha$ = 0.81).

**Intention.**    The measures of behavioral intention assessed how likely participants were to regularly engage in OHB, using a 7-point scale ranging from (1) extremely unlikely to (7) extremely likely. E.g., "I intend to brush my teeth twice a day by using toothpaste as described above in the next month on regular basis", "I will make an effort to brush my teeth twice a day

by using toothpaste as described above in the next month on regular basis" (1 = unlikely to 7 = likely). The total score ranges from 4 to 28 and a higher score indicates the higher the participant's readiness to perform OHB [39], ($\alpha = 0.9$).

**Data processing and analysis.** Data were entered into EpiData version 4.6 and exported into SPSS Version 26 for further data management and analysis. Cases having missed data in items measuring the theory of planned behavior was discarded. Variable coding and transformations were done to make the data set ready for analysis.

Descriptive analysis, the Student t-test, and correlation analyses were done using a statistical software package (SPSS 26, Inc., Chicago, IL, USA). SEM Analysis was also carried out using AMOS 23 (SPSS, Inc.) to confirm the existence of the proposed relationships among the constructs of TPB and to identify the most important predictor(s) of OHBI.

At the very beginning of the SEM analysis, the Kaiser-Meyer-Olkin (KMO) measure of sampling adequacy and Bartlett's test of sphericity was computed [40]. In addition to this, the multivariate normality test was done and the data was extremely departed from the multivariate normality assumption as a Mardias' coefficient was 50.6 [41]. Hence, the unweighted least squares (ULS) estimation technique was used [42].

The SEM analysis was done in two steps. In the first step, the assessment of the measurement model was done with a nine-factor CFA to assess the convergent and discriminant validity of the tool. Secondly, the eight-factor containing model was used to run the final SEM analysis to verify relationships and associations among exogenous, mediating, and endogenous variables. Misspecifications in the fitted model were assessed based on modification indices. Normed Fit Index (NFI), Adjusted Goodness of Fit Index (AGFI), Parsimony Normed Fit Index (PNFI), and Standardized Root Mean Square Residual SRMSR were used to assess the model fitness, the normal range of each index used in the present study depicted below NFI, PNFI, and AGFI of > 0.95, and SRMR of < 0.1 indicates good and acceptable model fitness respectively [43].

## Ethical issues

For this study, ethical clearance was obtained from the Institute Review Board of the University of Gondar a Ref. No: IPH/837/6/2012. Written consent was obtained from participants aged 18 and above. For participants with the age of less than 18, parental/guardian consent and assent from themselves was obtained. Moreover, permission letters and oral permission were obtained from the city education office and selected school principals respectively and each of the participants was included voluntarily. Indeed, the data were analysed anonymously.

## Results

### Sociodemographic results

A total of 393 students were involved in the study with a response rate of 97.5%. More than half of the participants were females (54%). The mean age of the participants was 18 (± 1.3) with the age range of 16 to 24. The majority (89.1%) of the participants were from public schools and more than half (51.7%) of them were grade 11 (Table 1).

### Oral hygiene knowledge and self-reported oral hygiene behavior

The mean OHK score of the respondents was found to be 6.74 (± 1.8) and it ranged from 0 to 11 (Table 2). Regarding participant's OHB, only 36 (9.2%), 81 (21%), and 67 (17%) of the respondents had brushed their teeth at least twice a day, cleaned their tongue, and between

**Table 1. Socio-demographic characteristics of the study participants (n = 393).**

| Variable | Response category | Frequency | Percent |
|---|---|---|---|
| Age | 16–20 | 373 | 94.9 |
| | 21–24 | 20 | 5.1 |
| Sex | Male | 181 | 46.1 |
| | Female | 212 | 53.9 |
| Marital Status | Single | 372 | 94.7 |
| | Married | 7 | 1.8 |
| | Widowed | 1 | .3 |
| | Engaged | 13 | 3.3 |
| Educational status of the participants | Grade 11 | 203 | 51.7 |
| | Grade 12 | 190 | 48.3 |
| The religion of the participants | Orthodox | 329 | 83.7 |
| | Muslim | 46 | 11.7 |
| | Protestant | 13 | 3.3 |
| | Catholic | 3 | .8 |
| | Other | 2 | .5 |
| Mother's occupation | Housewife | 213 | 54.2 |
| | Government employee | 106 | 27.0 |
| | Merchant | 46 | 11.7 |
| | NGO employee | 16 | 4.1 |
| | Farmer | 5 | 1.3 |
| | Other | 7 | 1.8 |
| Father's occupation | Government employee | 143 | 36.4 |
| | NGO employee | 72 | 18.3 |
| | Merchant | 134 | 34.1 |
| | Farmer | 24 | 6.1 |
| | Other | 20 | 5.1 |
| Mothers educational status | Unable to read and write | 47 | 12.0 |
| | Able to read and write | 90 | 22.9 |
| | Primary (1–8) | 38 | 9.7 |
| | Preparatory (9–12) | 92 | 23.4 |
| | Diploma and higher | 126 | 32.1 |
| Fathers educational status | Unable to read and write | 21 | 5.3 |
| | Able to read and write | 82 | 20.9 |
| | Primary (1–8) | 41 | 10.4 |
| | Secondary (9–12} | 79 | 20.1 |
| | Diploma and higher | 170 | 43.3 |
| With whom do you live? | With my parents | 315 | 80.2 |
| | With my siblings | 34 | 8.7 |
| | With my relatives | 16 | 4.1 |
| | Alone | 21 | 5.3 |
| | Other | 7 | 1.8 |
| School type | Government School | 350 | 89.1 |
| | Private School | 43 | 10.9 |

their teeth respectively. Each item was weighed and the sum score of OHB was computed [26]. The mean OHB score was about 7 (± 3.6). More than half (53%) of the participants had scored at or below the mean of OHB score (Table 3).

**Table 2. Oral hygiene knowledge of respondents by sex.**

| Items | Answer | Female (%) | Male (%) | Total (%) |
|---|---|---|---|---|
| For teeth maintenance, it matters how many times I eat sugary foods (biscuit, candy, chocolate…etc.) during a day. | Wrong | 5 | 10 | 7 |
| | Correct | 95 | 90 | 93 |
| To prevent caries, it is not enough to brush the crown covers only. | Wrong | 15 | 16 | 16 |
| | Correct | 85 | 84 | 84 |
| When brushing one's teeth it is important to put little pressure on the toothbrush. | Wrong | 11 | 19 | 15 |
| | Correct | 89 | 84 | 85 |
| To prevent dental caries, it is good to brush at least twice a day. | Wrong | 17 | 28 | 22 |
| | Correct | 83 | 72 | 78 |
| For tooth care, it doesn't matter if we use our toothbrush for a long time unless it is broken or lost. | Wrong | 83 | 77 | 80 |
| | Correct | 17 | 23 | 20 |
| Gum inflammation can disappear by itself. | Wrong | 83 | 83 | 83 |
| | Correct | 17 | 17 | 17 |
| Gum bleeding is a sign of periodontal disease. | Wrong | 29 | 34 | 32 |
| | Correct | 71 | 66 | 68 |
| In order to prevent gum inflammation, you also have to clean between your teeth. | Wrong | 18 | 32 | 25 |
| | Correct | 82 | 68 | 75 |
| Bad breath can be caused by gum disease. | Wrong | 33 | 38 | 35 |
| | Correct | 67 | 62 | 65 |
| Brushing before breakfast and before going to bed will enhance the preventive efficacy of tooth brushing. | Wrong | 19 | 33 | 25 |
| | Correct | 81 | 67 | 75 |
| When my gum does not bleed while brushing my teeth, there is nothing wrong with my gum | Wrong | 88 | 87 | 88 |
| | Correct | 12 | 13 | 12 |

### Theory of planned behavior variables

The mean score of all variables of TPB of the respondents was above the average (neutral). Moreover, the mean score of all most all variables of the theory was significantly higher among females except in indirect subjective norm and perceived behavioral control (Table 4).

### Correlation among TPB variables

Correlational analysis was done among the TPB variables, OHK, and OHB. In the analysis, all aforementioned variables exhibited a significant correlation with each other. However, as shown in the table below all variables showed the least correlation coefficient with OHK except indirect attitude (Table 5).

### Intention to oral hygiene behavior

The total score of the intention of the participants to oral hygiene behavior was extremely departed from normality. Hence, we computed the summary measure by using the median and inter-quartile range. Indeed, the median (interquartile range) intention towards OHB was 5.75 (4.5–7).

### Structural equation modelling analysis

Kaiser-Meyer-Olkin (KMO) sample adequacy test was 0.922 which supports the sample was adequate to proceed with factor analysis. In the meanwhile, Bartlett's test of sphericity was significant with p = .00, indicated that the correlation matrix among items was not an identity matrix [40].

**Table 3. Self-reported oral hygiene behavior of the participants (n = 393).**

| Items | Response category | Frequency | Percent |
|---|---|---|---|
| Frequency of tooth brushing | Not every day or not at all | 209 | 53.2 |
| | once a day | 148 | 37.7 |
| | twice a day | 36 | 9.2 |
| Moments of brushing | Don't brush every day or never at all | 209 | 53.2 |
| | Brush daily with any moment | 148 | 37.7 |
| | twice a day with no regular moments | 15 | 3.8 |
| | twice a day with any regular moments | 12 | 3.1 |
| | twice a day after breakfast and before going to bed | 9 | 2.3 |
| Amount of force used to brush | Forcefully | 213 | 54.2 |
| | Moderately | 180 | 45.8 |
| Changing toothbrush | Every one year or more | 173 | 44.0 |
| | Every six months | 67 | 17.0 |
| | Every three months | 153 | 38.9 |
| Duration of brushing | One minute or less | 84 | 21.4 |
| | Three minutes or more | 151 | 38.4 |
| | Two minutes | 158 | 40.2 |
| Toothpaste utilization | Not at all | 66 | 16.8 |
| | Sometimes | 61 | 15.5 |
| | Always | 266 | 67.7 |
| | Total | 393 | 100.0 |
| Interdental cleaning | Never | 122 | 31.0 |
| | Sometimes | 204 | 51.9 |
| | At least once a day | 67 | 17.0 |
| Tongue cleaning | Never | 128 | 32.6 |
| | Sometimes | 184 | 46.8 |
| | Everyday | 81 | 20.6 |
| | Total | 393 | 100.0 |

The proposed research model was composed of nine factors constructed based on TPB, OHK, and OHB. However, OHK was not included in the analysis because of the poor loading values of its items and hence, it didn't achieve a convergent validity. The final structural equation modeling analysis (SEM) showed acceptable model fit indices (Adjusted Goodness of fit

**Table 4. Descriptive statistics of TPB variables.**

| Variables | Total | | | | Female (n = 212) | | Male (n = 181) | |
|---|---|---|---|---|---|---|---|---|
| | Min | Max | Mean | SD | Mean | SD | Mean | SD |
| Direct attitude*** | 11 | 63 | 49.8 | 11.1 | 51.8 | 9.3 | 47.5 | 12.5 |
| Direct subjective norm* | 8 | 49 | 34.9 | 8.3 | 35.8 | 7.5 | 33.9 | 9.1 |
| Direct perceived control* | 4 | 28 | 21.2 | 5.6 | 21.7 | 5.0 | 20.6 | 6.2 |
| Indirect attitude** | 6 | 196 | 144.2 | 52.0 | 152.4 | 46.4 | 134.6 | 56.4 |
| Indirect subjective norm | 20 | 490 | 254.2 | 111.8 | 264.4 | 103.4 | 242.4 | 120.2 |
| Indirect perceived control | 10 | 245 | 129.8 | 60.3 | 128.6 | 57.3 | 131.3 | 63.7 |

*** significant at $p < 0.00$

** significant at $p < 0.01$

* significant at $p < 0.05$, Min minimum, Max Maximum.

**Table 5. Spearman's correlation among the TPB variables, OHK and OHB.**

| Variables | IPBC | IATT | ISN | DATT | DSN | DPBC | I | OHB | OHK |
|---|---|---|---|---|---|---|---|---|---|
| Indirect Perceived control (IPBC) | 1 | | | | | | | | |
| Indirect Attitude (IATT) | .335** | 1 | | | | | | | |
| Indirect Subjective Norm (ISN) | .448** | .502** | 1 | | | | | | |
| Direct Attitude (DATT) | .277** | .572** | .485** | 1 | | | | | |
| Direct Subjective Norm (DSN) | .298** | .481** | .614** | .600** | 1 | | | | |
| Direct Perceived behavioral control (DPBC) | .386** | .521** | .560** | .594** | .612** | 1 | | | |
| Intention (I) | .454** | .542** | .572** | .484** | .458** | .591** | 1 | | |
| Past Oral hygiene behavior (POHB) | .296** | .200** | .362** | .290** | .276** | .313** | .315** | 1 | |
| Oral Hygiene Knowledge (OHK) | .126* | .218** | .246** | .276** | .237** | .209** | .215** | .232** | 1 |

** Correlation was significant at the 0.01 level

* Correlation was significant at the 0.05 level.

index (AGFI = 0.984, NFI = 0.978, PNFI = 0.923, SRMR = 0.089), All of the fit indices indicated good model fit [44]. The aforementioned model fit indices results were achieved after freeing some covariances of measurement errors of the same construct (Fig 1).

As it is shown in Fig 1, the model explained a huge variance in oral hygiene behavioral intention as 66% of the variance intention and 72%, 69%, 72% of the variance in endogenous latent variables: direct ATT, SN, and PBC respectively was explained by the model.

## Association between direct and belief-based measures

Belief based measures (indirect ATT, SN and PBC) were significant predictors of their corresponding global measures (direct ATT, SN and PBC), (β = 0.85, p < 0.05), (β = 0.83, p < 0.05) and (β = 0.85, p < 0.01) respectively, indicating that the three beliefs (behavioral beliefs, normative beliefs and control beliefs) which were identified by elicitation study were adequately captured their corresponding overall measures (Table 6).

## Indirect predictors of behavioral intention

Belief based measurements of ATT, SN, and PBC were included in the SEM analysis as indirect predictors of intention via direct measures. Each (indirect ATT (β = 0.32, p < 0.01), SN (β = 0.28, p < 0.05) and PBC (β = 0.25, p< 0.05) of them was significantly and positively predicted intention indirectly (Table 6). Indirect ATT emerged as the strongest indirect predictor;

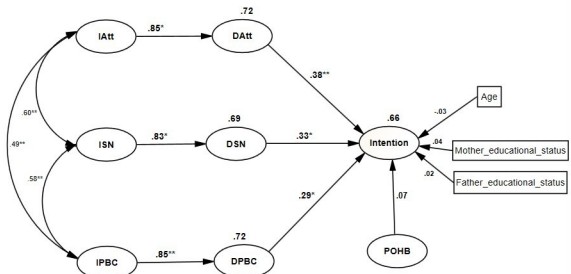

**Fig 1. A structural equation modeling analysis of OHBI based on the TPB framework.** ** p<0.01, *p < 0.05, IAtt = Indirect attitude, ISN = Indirect subjective norm, IPBC = Indirect perceived behavioral control, DAtt = Direct attitude, DSN = Direct subjective norm, DPBC = Direct perceived behavioral control, POHB = self-reported oral hygiene behavior.

**Table 6. Standardized regression weights of direct and indirect predictors of oral hygiene behavioral intention among preparatory school students of Gondar City, Northwest Ethiopia, 2020 (n = 393).**

| Direct predictors of OHBI | β | LB | UB | P value |
|---|---|---|---|---|
| Intention <---Direct subjective norm (DSN) | 0.33 | 0.05 | 0.83 | .02 |
| Intention <---Direct perceived behavioral control (DPBC) | 0.29 | 0.13 | 0.64 | .04 |
| Intention <---Direct attitude (DAT) | 0.38 | 0.21 | 0.64 | .003 |
| Intention <---Past oral hygiene Behavior (POHB) | 0.07 | 0.76 | 0.53 | .53 |
| Intention <---Mother's educational status | 0.04 | 0.07 | 0.15 | .44 |
| Intention <---Father's educational status | 0.02 | 0.08 | 0.12 | .71 |
| Intention <---Age of the participant | 0.03 | 0.15 | 0.07 | .54 |
| Indirect predictors of OHBI | | | | |
| Intention<---DSN<---indirect subjective norm | 0.28 | 0.04 | 0.74 | .023 |
| Intention <---DPBC <---indirect perceived control | 0.25 | 0.01 | 0.58 | .031 |
| Intention <---IAT<---Indirect attitude | 0.32 | 0.18 | .0.56 | .003 |

Note: LB lower border of 95% confidence interval, SE standard error, UB Upper border of 95% confidence interval, β = standardized path coefficient, <---direction of the effect, OHBI: Oral Hygiene Behavioral Intention.

indicating that student's OHBI was highly dominated by the value they gave to the possible outcomes of performing OHB and their evaluation of those outcomes ("avoid bad oral smell", "prevent dental caries", "keep my teeth beautiful" and "enables me to communicate with others freely"). However, students perceived pressure from their parents, siblings, close friends, and teachers about oral hygiene behavior and their perceived controls (time constraints, lack of aid materials, and fear of bad oral smell following discontinuation) were also play important indirect contributors to their OHBI.

Moreover, communicating with others like talking, laughing, and playing without feeling shame was the most important reason that the students used to value and evaluate OHB (β = 0.86, p < 0.01). Regarding significant others, students perceived that closest friends were the most important individuals who created positive social pressure on them to engage in oral hygiene behavior (β = 0.79, p < 0.01). Indeed, a lack of toothpaste was the most important control factor identified by the students (β = 0.81, p < 0.01).

## Direct predictors of intention

Direct ATT, SN and PBC, OHB, age and education of the participants, and paternal educational status were modelled directly to OHBI. All of the direct measures were significantly and positively predicted intention with a standardized path coefficient of 0.38, 0.33, and 0.29 for direct ATT, SN, and PBC (p < 0.05) respectively. This indicates that the higher the ATT towards OHB, the higher the positive social pressure from significant others, and the higher perceived power to control the barriers of oral hygiene behavior were significantly associated with the higher intention to improve oral hygiene behavior. In addition to this, direct ATT was found to be the most important predictor of OHBI. However, oral hygiene behavior OHB and some of the sociodemographic variables (age of the participant, maternal education, and paternal education) were not significant predictors of OHBI (Table 6).

## Discussion

In the present study, the determinants of intention to improve oral hygiene behavior was assessed. Both direct and indirect ATT, SN, and PBC were significant predictors of OHBI. The TPB provided acceptable model fit statistics and explained 66% of the variance in OHBI,

which indicates that the TPB has enough predictive utility in explaining OHBI [45]. This is in line with a study done in Norway where the model explained about 64% of the variance in intention [27]. However, it is higher when compared to a meta-analytic study [19] where the theory of planned behavior explained 39% of the variance in intention and to other studies conducted in Romania, Northern Ireland, and Indonesia where, 52%, 57.1%, and 57.6% of the variance in OHBI were explained by the model [24, 29, 31]. This discrepancy may be due to that in those previous studies, they were tried to predict intention with either of the indirect or direct measures of the model, unlike the current study where both measures were included in the analysis. Moreover, regression dilution may be another reason especially for the studies done in Northern Ireland and Indonesia by which their analysis was done using linear regression that doesn't account for measurement error, SEM.

In the present study, ATT, SN, and PBC were positively and significantly linked to OHBI, as supposed by the TPB. Meaning participants who had favourable ATT, strong positive social pressure from significant others, and higher perceived power to control over the barriers to OHB were found to had a stronger intention to improve OHB. This result is supportive of what is expected of in the TPB [18] and other TPB-based studies done in Ireland, Indonesia, Australia, and Dominican Republic [29–31, 46]. On the other hand, the findings of the present study are somewhat different from studies done in Romania, Canada, and Ireland where only ATT and PBC were significant predictors of OHBI [24, 28, 33]. The reason for such differences may be due to the variations in social economic and demographic variations across the study subjects. For example, the SN was not a significant factor in Canada and Australia, this may be due to higher individualization living style and low social support given to one another as compared to the current study participants living with strong social support lower individualism.

Moreover, the ATT emerged as the strongest predictor of OHBI, which implied that students had a stronger intention to improve oral hygiene behavior was mainly due to their belief concerning the importance of performing OHB and positive evaluation concerning the consequences of OHB. This result was in line with studies conducted in the Dominican Republic, Romania, and Iran [24, 29, 30]. This may be due to that human beings are rational, i.e., people perform a behavior if they believe that behavior is significant to them and evaluate its consequences the behavior positively, regardless of their residence. However, this result is inconsistent with studies done in Iran and Indonesia where the subjective norm was found to be the strongest predictor of OHBI [31, 34]. This may due to the socio-cultural difference among the study participants. For instance, all of the study participants of the study in Indonesia were Muslims, unlike the present studies where participants were followers of various religions.

The present study revealed that parental education did not play a critical role in determining the extent of intention to oral hygiene behavior among participants. In contrast, a previous study involving students has shown that higher parental education play a significant role in overall oral hygiene behavior [2]. It could be expected that more educated parents would be more aware of their children's oral health and more likely to influence them to engage in oral hygiene behavior. A possible explanation for the present finding may be that the participants in this sample were senior high school students, and thus parental influence probably plays less of a role than it does for younger students. Furthermore, self-reported oral hygiene behavior was also found to be an insignificant predictor but positively linked to oral hygiene behavior. This finding is contrasting to studies done in Romania and Ireland where self-reported oral hygiene behavior was a significant predictor of OHBI [24, 47]. This may be due to the proportion of students who performed the recommended oral hygiene behavior was very low as compared to the study done in Romania. In addition to this, the intention is ever-changing entity across time and event. For example, students may not at the right time to decide on their oral hygiene behavior or they may be overambitious of their future performance so that

the relationship between self-reported behavior and intention may not have strong correlation [48].

## Limitation of the study

The findings of this study should be interpreted with the following limitation, it didn't account for oral hygiene behavior to be predicted based on the theory variable which may show how much intention could be translated into the behavior. In addition to this, OHB was assessed by asking participants to recall and to report their experience in the past month, this might induce recall bias. Moreover, a study was conducted entirely based on the TPB which is an intrapersonal health behavior model where environmental, organizational, and policy-level factors were not considered.

## Strengths of the study

Notwithstanding these limitations, the present study has several implications. It provides support for the TPB in predicting oral hygiene behavioral intention and adds to a large body of literature that speaks to the efficacy of this model in the study area. Moreover, the strength of this study also includes that it accounts for the indirect predictors of OHBI which was measured by items constructed based on the accessible beliefs of the participants about OHB. This may give a hint for individuals or organizations who want to design oral hygiene promotion interventions by providing a focus of intervention. Indeed, the application of SEM is another strength of this study. In addition to this, this analysis technique takes measurement errors into account during analysis which is advantageous in the analysis containing latent variables such as TPB based data [49].

According to TPB, health behavior change is the result of the relationships between personal factors, and attributes of the behavior itself. People's attitudes, perceived social norms, and perceived control of the barriers/facilitators to perform a behavior affect behavioral intentions and actual performance of the behavior [17]. In this study, SEM analysis revealed the predictive strength of ATT, SN, and PBC for OHBI. Based on this analysis school oral hygiene interventions should give due emphasis to that attitudinal changes and consideration of beliefs regarding other people's support of the behavior. In addition to this, interventions should also target individuals' perceptions of behavioral control when seeking to promote OHB. An approach to enhancing an individual's control over engaging in OHB would be to make changes or intervene at the individual and environmental level. This may involve measures that increase the availability and accessibility of OHB aids particularly toothpaste and brush, for example, making such material to be free of tax so that students can access at a lower cost.

Furthermore, the impact of the COVID-19 pandemic may affect these findings. So, therefore, for the development of new oral health intervention, the so-called post-COVID-19 intervention, the previous results might give an indication. Moreover, during the COVID-19 pandemic, it is not only important to prevent becoming infected with the virus, but also to pay attention to daily personal hygiene activities, such as tooth brushing. And thus, to focus on promoting optimal oral health and to raise oral (self) care awareness among the public by oral health professionals is required [50].

## Conclusion

The TPB model explained a great deal of variance in students' intention to improve oral hygiene behavior, and all the TPB variables were positively and significantly linked to OHBI as proposed by the theory, indicated that the TPB showed adequate utility in predicting oral hygiene behavior in the study area. Furthermore, attitude towards oral hygiene behavior was

found to be the strongest predictor of intention to improve oral hygiene behavior. Though self-reported OHB was linked positively to OHBI, it was not found to be a significant predictor of student's intention to improve oral hygiene behavior.

## Recommendation

School-based oral hygiene behavior change interventions and/or researches will be benefited if they are guided by the theory of planned behavior. Moreover, such interventions should give due emphasis to attitudinal changes. Though addressing barriers of oral hygiene behavior and creating positive social pressure from significant others, also have an important role in enhancing students' intention to improve oral hygiene behavior.

## Supporting information

**S1 Fig. Diagrammatic representation of the conceptual framework based on theory of planned behavior and different literatures [25, 38, 51, 52].**
(TIF)

## Acknowledgments

We would like to acknowledge the University of Gondar for funding this research project. Moreover, we would like to express our gratitude to the Gondar city education staff office for their cooperation and provision of permission to conduct this study. Indeed, we are also grateful to mention our thanks to the study participants for their time and willingness to participate.

## Author Contributions

**Conceptualization:** Kegnie Shitu.

**Data curation:** Kegnie Shitu.

**Formal analysis:** Kegnie Shitu.

**Funding acquisition:** Kegnie Shitu.

**Investigation:** Kegnie Shitu.

**Methodology:** Kegnie Shitu, Yvonne A. B. Buunk-Werkhoven, Simegnew Handebo.

**Resources:** Yvonne A. B. Buunk-Werkhoven.

**Software:** Kegnie Shitu.

**Supervision:** Mekuriaw Alemayehu, Yvonne A. B. Buunk-Werkhoven, Simegnew Handebo.

**Validation:** Kegnie Shitu, Mekuriaw Alemayehu.

**Writing – original draft:** Kegnie Shitu.

**Writing – review & editing:** Mekuriaw Alemayehu, Yvonne A. B. Buunk-Werkhoven, Simegnew Handebo.

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
