## [Decision Letter · Decision Letter 0]

2 Dec 2020

PONE-D-20-32078

Determinants of Intention to Improve Oral Hygiene Behavior Among Students Based on the Theory of Planned Behavior: A Structural Equation Modelling Analysis

PLOS ONE

Dear Dr. Shitu,

Thank you for submitting your manuscript to PLOS ONE. After careful consideration, we feel that it has merit but does not fully meet PLOS ONE’s publication criteria as it currently stands. Therefore, we invite you to submit a revised version of the manuscript that addresses the points raised during the review process.

We look forward to receiving your revised manuscript.

Kind regards,

Mohammad Asghari Jafarabadi

Academic Editor

PLOS ONE

Reviewers' comments:

Reviewer's Responses to Questions

**Comments to the Author**

1. Is the manuscript technically sound, and do the data support the conclusions?

Reviewer #1: Partly

Reviewer #2: Yes

Reviewer #3: Yes

2. Has the statistical analysis been performed appropriately and rigorously? 

Reviewer #1: No

Reviewer #2: Yes

Reviewer #3: Yes

3. Have the authors made all data underlying the findings in their manuscript fully available?

Reviewer #1: No

Reviewer #2: Yes

Reviewer #3: Yes

4. Is the manuscript presented in an intelligible fashion and written in standard English?

Reviewer #1: No

Reviewer #2: Yes

Reviewer #3: Yes

5. Review Comments to the Author

Reviewer #1: This manuscript is to determine/predict the behavior of students based on the theory of planned behavior (i. e, to predict an individuals intension to engage in a behavior at a specific time and place). It also intends to use the structural equation modelling analysis to show the structural relationship by using a combination of factor analysis and multiple regression, which basically should measure the relationship between the measured variables and the latent construct.

Though, the topic seem an interesting one, the manuscript needs to be worked on by authors before it could be considered by this journal. 1. there is the need for the manuscript to be edited by a native speaker since there is a lot of grammatical issues and typographical errors, for example, see line 107 where you typed 2029/2020.

2. Introduction

please include literature on how the TPB have been used to determine/predict behaviors of people in other parts of the world before you limit it to Africa, and then Eastern Africa and then to Ethiopia before you justify your topic. As it stands now, the later part of the introduction is more about the explanation of the TPB which will only need about a line in the manuscript.

3. methods

This is where i find most statistical flaws. Please explain further and clearly on how the sample size was arrived rather than the vague equations. Was it calculated on assumption( because of the 50%), if 'yes' that will mean there have been no study conducted so far on this area as far as Ethiopia is concerned. If 'No' then please cite the paper upon which you are basing your calculations with the sample that those authors used.

Please show your exposure and dependent variables and how they will be measured on the methodology section.

4. Results

Again, I would recommend that you do not categorize your variables, so that you can explore your data. And this is where you would have to explore more, for example on socio-demographics, please show how many were 15 years and so on. Then report on Means, median, Standard deviations and so on ( please read a similar manuscript like 'Åstrøm AN, Lie SA, Gülcan F. Applying the theory of planned behavior to self-report dental attendance in Norwegian adults through structural equation modelling approach. BMC Oral Health. 2018') and report appropriately if you want to stick to the the TPB.

Again on your results i do not see how those variables you measured in Table 1 determines/ predicts the students behavior as per the theory, for example how does 'Mother's educational status determine/predicts the oral hygiene of the student'. I also do not see any figure that is displaying the relationship between the regression and the variables. This is why you will need to redo the analysis so that you can well display the relationships between the Means Medians, SD and others with your multiple regression. That will bring out the beauty and understanding of what you want people to know.

5. Discussion

When you are done with the above, that would affect your discussions, and I would like to see how your work differs from others rather than just conforming to what is already done. Please discuss your own work for example on how ' mother's education determines the behavior of a child, the reasons behind that and many more. I strongly believe that when this is redone, you would have a lot to discuss looking at your variables, and your paper would stand out among the lots.

Good Luck!

Reviewer #2: Determinants of Intention to Improve Oral Hygiene Behavior Among Students Based on the Theory of Planned Behavior: A Structural Equation Modelling Analysis

1- No extensive details are needed in study area (delete lines 98 to 101)

2- Check the academic year 2029/2020 in line 107

3- In exclusion criteria, why students transferred in the academic year not included in the study?

4- Did all students accept to participate in the present study??

5- How long did data collection take time? It should be mentioned in data collection section

6- Items of knowledge (OHK) should be described clearly in the data collection tool

7- Describe the eight items concerning OHB in the data collection tool

8- Line 179, 6 should be replaced by 7

9- Add table describing oral hygiene knowledge and self-reported oral hygiene behavior.

10- The discussion did not involve studies from developing countries

11- In the acknowledgement, you should acknowledge study participants (students).

Reviewer #3: A well written manuscript with intelligent inferences drawn from the statistical analysis. The methodology is sound and the research area is of importance in its relevant field. Behavioural interventions have been increasingly recognized as effective preventive strategies in different domains of health.

There were few grammatical mistakes detected in various components and a professional proof reading of manuscript is advised.

6. PLOS authors have the option to publish the peer review history of their article (what does this mean?). If published, this will include your full peer review and any attached files.

Reviewer #1: No

Reviewer #2: No

Reviewer #3: No

---

## [Author Response · Author response to Decision Letter 0]

21 Jan 2021

Dear Mohammad Asghari Jafarabadi, editor and reviewers:

Hereby, we resubmit the enclosed –revised- manuscript ID PONE-D-20-32078 -[EMID:a6efc58a6a98f997], that is entitled “Determinants of Intention to Improve Oral Hygiene Behavior among Students Based on the Theory of Planned Behavior: A Structural Equation Modelling Analysis” to your journal.

We have read the comments carefully and we were able to implement all of them (see manuscript with track changes). While reading the manuscript critically, we spotted English grammar errors, which we and a language expert have corrected too. We hope that our revision will be felt as an improvement. We certainly feel this manuscript has improved thanks to the suggestions of the reviewers.

The datasets generated and/or analyzed in the current study are available at University of Gondar, College of Medicine and Health Science, Institute of Public Health in hard and soft copy repository [www.UoG.edu.et]. 

Response to reviewer 1 comments and suggestions

This manuscript is to determine/predict the behavior of students based on the theory of planned behavior (i.e., to predict an individual’s intention to engage in a behavior at a specific time and place). It also intends to use the structural equation modelling analysis to show the structural relationship by using a combination of factor analysis and multiple regressions, which basically should measure the relationship between the measured variables and the latent construct. Though, the topic seems an interesting one, the manuscript needs to be worked on by authors before it could be considered by this journal. 

1. There is the need for the manuscript to be edited by a native speaker since there is a lot of grammatical issues and typographical errors, for example, see line 107 where you typed 2029/2020

Response: Thanks for this comment. We gave this manuscript to an English language expert, Mr. Fikadie, for an English language edition. Then the entire manuscript was edited for grammatical and typographical errors based on his suggestion and commentaries. And of course, 2029/2020 must be 2019/2020. Thank you.

2. Introduction: please include literature on how the TPB has been used to determine/predict behaviors of people in other parts of the world before you limit it to Africa, and then Eastern Africa, and then to Ethiopia before you justify your topic. As it stands now, the latter part of the introduction is more about the explanation of the TPB which will only need about a line in the manuscript.

Response: It is clear that the theory of planned behavior has been applied to the prediction of the range of behaviors/intentions. However, our initial objective was to find evidence that demonstrates the usefulness of theory in predicting oral hygiene behavior. In this regard, we did not find any published papers involving the application of TPB in the prediction of oral health behavior of students in Ethiopia, not even in Africa. For this reason, we did not describe the usefulness of TPB in Ethiopia or Africa. However, we have now added some evidence to demonstrate how useful TPB is in predicting different health behaviors other than oral hygiene behaviors in Ethiopia. We also tried to mention the usefulness of TPB in predicting various health behaviors worldwide. (See manuscript, line 67 to 79)

3. Methods: This is where i find most statistical flaws. Please explain further and clearly on how the sample size was arrived rather than the vague equations. Was it calculated on assumption (because of the 50%), if 'yes' that will mean there have been no study conducted so far on this area as far as Ethiopia is concerned. If 'No' then please cite the paper upon which you are basing your calculations with the sample that those authors used. Please show your exposure and dependent variables and how they will be measured on the methodology section.

Response: Unlike the sample sizes for means and proportions, which are obtained with a certain level of confidence from mathematical calculations, the sample sizes for the estimation of SEM are derived in another way. There is no single all fitted sample size calculation for SEM-based studies. As a result, there are various recommendations. However, it is taught that the required sample size for SEM depends on the complexity of the model, the relationship between the observed and latent variables, and the type I and II error. Some scholars use the rule of thumb and others use a kind of hypothesis to compute a sample for the SEM-based study. For our study, the sample size was calculated using a statistical calculator developed by Professor Daniel. This calculator calculates the required sample size as per the following assumptions.

• The number of observed and latent variables in the model, 

• The anticipated effect size and 

• The desired probability, and statistical power levels. 

Reference: Soper, D.S. (2020). A-priori Sample Size Calculator for Structural Equation Models [Software]. Available from https://www.danielsoper.com/statcalc

We also mentioned the study sample size and number of variables used in the present study (see in the manuscript, line 118-142). Furthermore, how we measured each variable is already described in the “Data Collection and Measurement” sub-heading of the Methodology section. (See in the manuscript in line 143-234)

 4. Results: Again, I would recommend that you do not categorize your variables so that you can explore your data. And this is where you would have to explore more, for example on socio-demographics, please show how many were 15 years and so on. Then report on Means, median, Standard deviations, and so on (please, read a similar manuscript like 'Åstrøm AN, Lie SA, Gülcan F. Applying the theory of planned behavior to self-report dental attendance in Norwegian adults through a structural equation modeling approach. BMC Oral Health. 2018') and report appropriately if you want to stick to the TPB.

Again on your results, I do not see how those variables you measured in Table 1 determines/predict the student's behavior as per the theory, for example how does 'Mother's educational status determine/predicts the oral hygiene of the student'. I also do not see any figure that is displaying the relationship between the regression and the variables. This is why you will need to redo the analysis so that you can well display the relationships between the Means Medians, SD, and others with your multiple regression. That will bring out the beauty and understanding of what you want people to know.

Response: The age range of the students is very narrow (16-24 years). Therefore, we did not rank the variable in more categories than we did. We have already mentioned the mean and standard deviation of the age of participants (see line 268 of the manuscript). Furthermore, the primary objective of this study was not to predict students' oral hygiene behavior, but their intention to engage in OHB. According to the TPB, the behavior should be measured after one week, one month, or three months, and so on depending on the nature of the behavior. However, as it is mentioned in the limitation section of our study, behavior and other TPB variables were measured at the same time. As a result, we did not consider past OHB as an outcome variable. Nevertheless, we used this variable as a predictor of the intention to engage in OHB in the future. 

Moreover, we did an SEM analysis to confirm the hypothesized relationships supposed by the TPB. The extent to which one variable has an impact over the other variable is measured by the path coefficient that links those variables together. This estimate is analogous to a regression coefficient in the ordinary regression analysis. Moreover, the significance of the association is determined by the critical ratio and p-value of the estimate. All these things might be presented in diagrams or tables. In the previous manuscript, we presented the SEM result in diagrammatic form. In the revised manuscript, we have added a table containing the direct and indirect path coefficient with a 95% confidence interval to make it easier to understand. (See the manuscript in table 6, line 355) 

5.Discussion: When you are done with the above, that would affect your discussions, and I would like to see how your work differs from others rather than just conforming to what is already done. Please discuss your own work for example on how ' mother's education determines the behavior of a child, the reasons behind that and many more. I strongly believe that when this is redone, you would have a lot to discuss looking at your variables, and your paper would stand out among the lots.

Response: We have made many changes to the discussion section of this manuscript. In particular, we discussed variables that were not significant predictors of OHBI in our study, but in earlier studies (see line 398-413 of the manuscript). We have tried to compare and contrast our results to what has already been done and provide possible explanations for conformities as well as differences. The current study is also the first to assess oral hygiene behavioral intention and its determinants among students in our country, where oral health practices are very poor in this population group. In addition to this, the study identified the most important psychosocial determinant of intention to perform OHB which is of paramount importance for behavioral change intervention aimed at improving the OHB of students. In the end, improve their oral and general health. 

Response to reviewer 2 comments and suggestions

1- No extensive details are needed in the study area (delete lines 98 to 101)

Response: Thank you, we have deleted these sentences (see in manuscript with track changes in lines 115-118)

2- Check the academic year 2029/2020 in line 107

Response: Sorry for this mistake, we have changed this in 2019/2020 (See in the manuscript in line 107)

3- In exclusion criteria, why students transferred in the academic year not included in the study?

Response: We excluded students who were transferred within the school year at the time the data was collected. We have done this due to the behavior/intention towards oral hygiene varies according to residence and culture in our facility. For example, oral hygiene behavior is extremely low in rural residents compared with the urban population. Furthermore, according to Planned Behavior Theory (BPT), the intention of the student is determined by the student's attitude, subjective standard, and perceived behavioral control of oral hygiene behavior. In this respect, students from different societies may have different attitudes, subjective norms, perceived control, and intentions which may affect the associations between the TPB variable. Moreover, these students are not well adapted to the social system i.e. they may act as what they behaved in their origin or may overact because of over-expectation of living in cities, both affects our outcome of interest. From this point of view, it is difficult to draw any conclusion from data containing information from students who are not representative of the city's students. Thus, we excluded those students from the study to avoid under /overestimation of the study’s result.

4- Did all students accept to participate in the present study?? 

Response: No. As we have mentioned in the result section of the abstract and main text, 393 (97.5%) of the students participated from the estimated sample size i.e. 403. (See the manuscript in line 34 and 267) 

5- How long did data collection take time? It should be mentioned in the data collection section

Response: Alright, we have mentioned the data collection period in line 144 in the manuscript under the data collection section.

6- Items of knowledge (OHK) should be described clearly in the data collection tool

Response: All items used to measure oral hygiene knowledge are depicted in the result section (see Table 2 in line 280-281 in the manuscript). Thus, we didn’t list those items in the data collection tool just not to make things redundant. 

7- Describe the eight items concerning OHB in the data collection tool

Response: All items concerning OHB are depicted in the result section (see in table 3 in line 282-283 in the manuscript). Thus, we didn’t list those items in the data collection tool just not to make things redundant. 

8- Line 179, 6 should be replaced by 7

Response: It was a typological error we made and correction was made accordingly (see in line 216 in the revised manuscript with track changes)

9- Add table describing oral hygiene knowledge and self-reported oral hygiene behavior.

Response: A table describing oral hygiene knowledge and self-reported oral hygiene behavior was added in the result section (see the revised manuscript with track change in line 310 (table 2) and line 312 (table 3) respectively).

10- The discussion did not involve studies from developing countries

Response: Yeah, you are right. That was also our concern. Unfortunately, to the best of our knowledge, there are no published studies concerning the determinants of intention to oral hygiene behavior in those countries.

11- In the acknowledgment, you should acknowledge study participants (students).

Response: Thank you. Correction was made upon your comment (See revised manuscript with track changes in line 511 to 513).

Response to Reviewer 3 comments and Suggestions

A well written manuscript with intelligent inferences drawn from the statistical analysis. The methodology is sound and the research area is of importance in its relevant field. Behavioural interventions have been increasingly recognized as effective preventive strategies in different domains of health. There were few grammatical mistakes detected in various components and a professional proof reading of manuscript is advised.

Response: Thank you. The manuscript reviewed for language by an English expert and revisions were made accordingly.

---

## [Decision Letter · Decision Letter 1]

1 Feb 2021

Determinants of Intention to Improve Oral Hygiene Behavior Among Students Based on the Theory of Planned Behavior: A Structural Equation Modelling Analysis

PONE-D-20-32078R1

Dear Dr. Shitu,

We’re pleased to inform you that your manuscript has been judged scientifically suitable for publication and will be formally accepted for publication once it meets all outstanding technical requirements.

Kind regards,

Mohammad Asghari Jafarabadi

Academic Editor

PLOS ONE

Reviewer's Responses to Questions

**Comments to the Author**

1. If the authors have adequately addressed your comments raised in a previous round of review and you feel that this manuscript is now acceptable for publication, you may indicate that here to bypass the “Comments to the Author” section, enter your conflict of interest statement in the “Confidential to Editor” section, and submit your "Accept" recommendation.

Reviewer #1: All comments have been addressed

Reviewer #2: All comments have been addressed

2. Is the manuscript technically sound, and do the data support the conclusions?

Reviewer #1: Yes

Reviewer #2: Yes

3. Has the statistical analysis been performed appropriately and rigorously? 

Reviewer #1: Yes

Reviewer #2: Yes

4. Have the authors made all data underlying the findings in their manuscript fully available?

Reviewer #1: Yes

Reviewer #2: Yes

5. Is the manuscript presented in an intelligible fashion and written in standard English?

Reviewer #1: No

Reviewer #2: Yes

6. Review Comments to the Author

Reviewer #1: Raised comments have been addressed, Manuscripts technically and scientifically sounds good, and results have been appropriately conducted. But, I still think you can do more with the language editing.

Reviewer #2: Determinants of Intention to Improve Oral Hygiene Behavior Among Students Based on the Theory of Planned Behavior: A Structural Equation Modelling Analysis

Thanks for addressing the comments . All the comments have been addressed carefully

no other comments

7. PLOS authors have the option to publish the peer review history of their article (what does this mean?). If published, this will include your full peer review and any attached files.

Reviewer #1: No

Reviewer #2: No

---

## [Editor Report · Acceptance letter]

8 Feb 2021

PONE-D-20-32078R1 

Determinants of Intention to Improve Oral Hygiene Behavior Among Students Based on the Theory of Planned Behavior: A Structural Equation Modelling Analysis 

Dear Dr. Shitu:

I'm pleased to inform you that your manuscript has been deemed suitable for publication in PLOS ONE. Congratulations! Your manuscript is now with our production department. 

Kind regards, 

on behalf of

Professor Mohammad Asghari Jafarabadi 

Academic Editor

PLOS ONE